# Multi-Kingdom Gut Microbiome Interaction Characteristics Predict Immune Checkpoint Inhibitor Efficacy Across Pan-Cancer Cohorts

**DOI:** 10.3390/microorganisms13112595

**Published:** 2025-11-14

**Authors:** Tong Qiao, Zhenjun Zhu

**Affiliations:** 1Department of Clinical Medicine & Department of Food Science and Engineering, Jinan University, Guangzhou 510632, China; 13193810515@163.com; 2State Key Laboratory of Food Science and Resources, Nanchang University, Nanchang 330027, China

**Keywords:** multi-kingdom microbiome, ICI therapy, WSNF, co-occurrence networks, metagenomic sequencing

## Abstract

An increasing number of studies have confirmed that the gut microbiota, especially bacteria, is closely related to the efficacy of immune checkpoint inhibitor (ICI) therapy. However, the effectiveness of multi-kingdom microbiota and their interactions in predicting the therapeutic effect of ICI therapy remains uncertain. We integrated extensive gut metagenomic databases, including 1712 samples of 10 cohorts from 7 countries worldwide, to conduct rigorous differential analysis and co-occurrence network analysis targeting multi-kingdom microbiota (bacteria, fungi, archaea, and virus). We ultimately identified two subtypes (C1 and C2) by employing a weighted similarity network fusion (WSNF) method. Subtype C2 exhibited higher microbial diversity, better treatment response, and improved prognosis compared to subtype C1. Notably, subtype C2 was associated with higher abundance of beneficial genera such as *Bacteroides* and *Kluyveromyces*, while subtype C1 contained potentially detrimental taxa like *Malassezia*. A multi-kingdom model incorporating 32 genera demonstrated superior predictive accuracy for ICI therapy efficacy compared to single-kingdom models. Co-occurrence network analysis revealed a more robust and interconnected microbiome in subtype C2, suggesting a stable gut environment correlates with effective ICI therapy efficacy. This study highlights the potential of a multi-kingdom signature in predicting the efficacy of ICI therapy, offering a novel perspective for personalized therapy in oncology.

## 1. Introduction

Immune checkpoint inhibitor (ICI) therapy, including agents targeting programmed cell death protein 1 (PD-1)/programmed cell death ligand 1 (PD-L1), cytotoxic T-lymphocyte-associated protein 4 (CTLA-4), etc., has demonstrated notable clinical efficacy across multiple cancer types [1,2,3,4] and represents a revolutionary advancement in cancer treatment. However, resistance to ICI therapy is frequently observed, with roughly 55% of melanoma patients exhibiting innate resistance to single-agent PD-1 inhibitor [5]. Additionally, the incidence of immune-related adverse events (irAEs) during treatment remains high. For instance, nearly all (>95%) patients with advanced melanoma receiving combined CTLA-4 and PD-1 therapy experience some degree of irAEs [6]. Therefore, finding strategies to enhance the efficacy of ICI therapy is a pressing clinical challenge that needs to be fully addressed.

Gut microbial dysbiosis signatures have been found in cancer patients, which are closely related to the response to ICI therapy and can accurately predict efficacy [7,8]. Several studies have reported changes in the gut microbiome of cancer patients receiving ICI therapy and have identified certain crucial bacterial species. For instance, one study identified that *Ruminococcaceae SGB15234* and *Blautia SGB4831* closely correlate better responses to ICI therapy in melanoma [9], while another study identified key bacterial genera *Akkermansia* were associated with improved responses in lung cancer patients receiving ICI therapy [10]. Furthermore, clinical trials have validated that fecal microbiome transplantation (FMT) can effectively improve ICI efficacy by alleviating dysbiosis in the gut [11]. Therefore, the identification of unique gut microbiota in cancer patients undergoing ICI therapy for prediction and intervention represents a promising therapeutic strategy.

Previous studies on the gut microbiome, especially those exploring its link with ICI therapy, have typically focused exclusively on bacterial species. Non-bacterial microorganisms, including fungi, viruses and archaea have also undergone significant alterations in various cancers, adding to the complexity of microbiome association studies. For instance, *Candida albicans* show significant enrichment in the lesions of oral squamous cell carcinoma and exhibit a positive correlation with various stages of tumor progression, suggesting their potential influence on cancer development and aggressiveness [12]. Additionally, the *Malassezia* genus has been shown to significantly promote the development of pancreatic ductal adenocarcinoma [13]. Moreover, the Epstein–Barr virus is detected in approximately 9% of gastric cancers, highlighting its involvement in the pathogenesis of this malignancy [14]. Furthermore, Cocker et al. demonstrated the potential application of *Halophilic archaea* as diagnostic biomarkers for colorectal cancer (CRC), while also elucidating the interactions between archaea and bacteria that are enriched in CRC and their contributions to the pathogenesis of colon cancer [15]. The above studies highlight the important roles of the multi-kingdom microbiome in cancer. It remains unknown whether characteristic patterns also exist among other kingdoms, such as fungi, archaea, and viruses, and whether incorporating multi-kingdom microbial profiles could enhance the prediction accuracy of ICI therapy. Furthermore, the complexity of interactions among these communities has yet to be fully elucidated.

In this study, we analyzed fecal metagenomic data from 1712 patients undergoing ICI therapy across 7 countries worldwide. Then, we identified two subtypes (C1 and C2) that could significantly distinguish ICI therapy efficacy by WSNF clustering. We found that subtype C2 exhibited higher microbial diversity across the different kingdoms. The multi-kingdom model predicts the efficacy of ICI therapy more accurately than the single-kingdom model. Network analysis showed that subtype C2 had a stable and diverse gut microbiome community. This study utilized a metagenomics approach to characterize the complex signatures of the gut microbiome and explore its effectiveness in mediating ICI therapy efficacy.

## 2. Materials and Methods

### 2.1. Study Cohorts

We included a total of 1712 samples of 10 cohorts from 7 countries. These samples are fecal specimens that underwent metagenomic sequencing. All patients received ICI therapy (Table 1). Sequencing data of the populations from USA (PRJNA762360 [16], PRJNA541981 [17], PRJEB22893 [18], PRJNA399742 [19], PRJNA770295 [20], melanoma), France (PRJNA751792 [21], NSCLC; PRJEB22863 [22], NSCLC and RCC), Germany (PRJEB54704 [23], B cell lymphoma), Netherlands, Spain, and UK (PRJEB43119 [24], melanoma) and China (PRJNA615114 [25], Gastrointestinal Cancer) were included in this study. Metadata were manually curated from the published papers. The datasets used in this study can be accessed through the NCBI database using the provided SRA study numbers to obtain the raw metagenomic databases.

### 2.2. Metagenomic Analysis

A series of key procedures were implemented to standardize datasets from different sequencing platforms. Raw data were subjected to quality control using FastQC, which assessed sequence quality metrics, including base quality scores and sequence length distribution. Subsequently, Trimmomatic was employed for trimming and filtering the sequences, involving the removal of low-quality bases, adapter sequences, and sequences shorter than a specified threshold. Host DNA contamination was addressed by aligning the sequences against the human genome (GRCh37) and removing those counts identified as host-derived. To mitigate batch effects, we employed a rarefaction step to standardize the sequencing depth across samples to 915968 using the “rrarefy” function in R package vegan (version 2.6-10).

Taxonomic classification of bacteria, archaea, fungi, and viruses was assigned to metagenomic reads using Kraken2 (version 2.1.1), a metagenomic taxonomy classifier that utilizes k-mer-based algorithms [26]. A custom database consisting of 8108 bacterial, 432 archaeal, 6658 viral, and 432 fungal reference genomes was obtained from the NCBI RefSeq database. Each k-mer was mapped to the lowest common ancestor of all reference genomes with exact k-mer matches. Then, each query was classified as a specific taxon with the highest total k-mer hits matched by pruning the general taxonomic trees affiliated with the mapped genomes. Bracken (version 2.5.0) was used to accurately estimate taxonomic abundance based on Kraken2 [27]. The read counts of species were converted into relative abundance for further analysis.

### 2.3. Weighted Similarity Network Fusion (WSNF) Analysis

The weighted similarity network fusion (WSNF) approach was employed to effectively integrate community structure information from four distinct microbial domains: archaea, bacteria, fungi, and viruses, in order to conduct comprehensive sample clustering analysis. To begin with, the relative abundance data for each microbial community were subjected to Hellinger transformation. This transformation is a critical step designed to mitigate potential biases arising from data sparsity, ensuring that subsequent analyses are more robust and reflective of true community dynamics. Following the transformation, Bray–Curtis distance matrices were calculated for each microbial domain based on pairwise sample comparisons. These distance matrices were then converted into similarity matrices, capturing the relationships between different samples in each domain. Recognizing the variability in sample sizes across these microbial domains, weights were assigned to each similarity matrix appropriately. This weighting ensures a balanced contribution from each domain when merging the data. During the fusion process, a customized WSNF algorithm iteratively updated the integrated similarity network, maintaining matrix symmetry and normalization. This careful maintenance is essential for ensuring the stability and reliability of the fusion. To determine the optimal fusion parameters, as well as the ideal number of clusters, multiple iterations were conducted to evaluate silhouette scores. These scores provide an insight into the effectiveness of each clustering solution and were calculated using a silhouette function implemented in Python (version 3.12.7) [28].

### 2.4. Microbial Diversity Analysis

Alpha diversity was evaluated using both the Shannon and observed species indices, providing insights into the overall microbiome diversity as well as diversity within individual kingdoms. This analysis was conducted using the R package phyloseq (version 1.44.0). To assess the differences between identified clusters, *t*-tests were performed, which allowed for the comparison of mean diversity metrics across clusters. In addition, beta diversity was estimated utilizing Bray–Curtis and Jaccard dissimilarities, offering a comparative measure of the overall microbiome and each kingdom’s dissimilarity. To examine differences between clusters in beta diversity, the PERMANOVA statistical test was employed, ensuring a robust analysis of community structure variations.

### 2.5. Survival Analysis

Survival curves were estimated using the Kaplan–Meier method, a widely used statistical procedure for analyzing the time until an event occurs, such as death or disease progression. The analysis incorporated the log-rank test, enabling the assessment of the significance of differences observed in survival curves. To conduct these analyses, the R programming language was utilized, specifically employing the R package survival (version 3.8-3) and survminer (version 0.5.0).

### 2.6. Differential Taxa Analysis and Machine Learing

From each cluster, 70% of samples were randomly selected for differential analysis to ensure a representative distribution for the examination of microbial communities. Considering the sparse distribution of the microbiome, we utilized the DESeq2 package (version 1.40.2), enabling us to conduct inter-cluster differential analysis. We identified the most significantly differentiated abundant species within each kingdom, which were employed as diagnostic models for distinguishing between the two clusters. To determine significance, we established a threshold with *P*.adj of less than 0.05, applying the Benjamini–Hochberg correction to control for false discovery rates. Additionally, we set thresholds for |log_2_FC| to determine significant species: 1 for bacteria, 0.5 for fungi and archaea, and 0.25 for viruses. The remaining 30% of the samples constituted an independent test set for machine learning validation. For the machine learning phase, we implemented random forest and support vector machine algorithms to compare the classification capabilities of multi-kingdom diagnostic models against single-kingdom models. To minimize noise and ensure robustness, we employed 10-fold cross-validation. Finally, the performance of the diagnostic models was assessed using receiver operating characteristic (ROC) curves, generated through the R package pROC (version 1.18.5), enabling us to evaluate the models’ sensitivity and specificity comprehensively.

### 2.7. Microbial Community Network Analysis

We utilized the command-line tool FastSpar (https://github.com/scwatts/fastspar) to perform SparCC correlation analyses on the identified differential genera from our samples. This method facilitated the examination of correlations among the microbial community structures by accounting for the compositional nature of microbiome data. To ensure the reliability of our results, we conducted 500 bootstrap iterations, which allowed us to derive precise *p*-values for the correlations observed. For statistical significance, we established a threshold of *P*.adj less than 0.05. Following the correlation analysis, we proceeded to visualize the identified “microbial communities” within different clusters using the software Gephi (version 0.10). Gephi is an effective tool for graphical visualization, enabling us to create informative network representations of the microbial correlations.

### 2.8. Statistical Analysis

The *t*-test was used to evaluate whether continuous variables differ significantly between groups and the chi-square test for categorical variables. PERMANOVA was performed on variables derived from distance matrices. *p* < 0.05 or 0.01 or 0.001 is considered statistically significant.

## 3. Results

### 3.1. Multi-Kingdom Microbial Clustering Using WSNF Method

To evaluate the gut microbial characteristics of patients receiving ICI therapy, we retrieved published metagenomic databases from 1712 samples of 10 cohorts across 7 countries worldwide (Figure 1 and Table 1). All patients had received ICI therapy, and the majority had received anti-PD-1 treatment.

**Figure 1 microorganisms-13-02595-f001:**
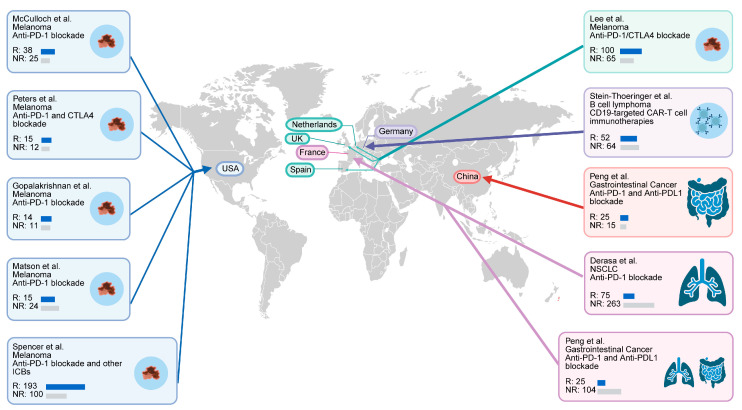
Multiple cohorts spanning several countries used in this study. Global map representing 1712 samples. Fecal metagenomes were retrieved from 10 published cohorts in 7 countries [16,17,18,19,20,21,22,23,24,25].

Additionally, samples were screened based on the criteria that the microbial phylotypes had a prevalence of at least 10% in the patient cohort and that the absolute number of species was greater than 300. Finally, 965 samples were retained, including those from patients with melanoma *(n* = 400), non-small cell lung cancer (*n* = 428), gastrointestinal tumors (*n* = 39), and renal cell carcinoma (*n* = 98). It was statistically found that the number of bacteria (5057) > archaea (166) > fungi (54) > virus (22) in all samples. The WSNF method was employed to integrate the abundance data for bacteria, fungi, viruses, and archaea. The weighted similarity matrix was analyzed to spectral cluster; ultimately, the patients were assigned to two clusters (subtype C1/subtype C2, 524/441) (Figure 2A,B). A similarity matrix was calculated using Bray–Curtis distances for each kingdom with weights assigned based on microbial counts, some of which showing significant differences (Figure 2C).

### 3.2. Analysis of Microbial Diversity and Its Correlation with ICI Therapy Outcomes in Patient Subtypes

We conducted the microbial diversity analysis at the species level between the subtypes. The results showed that there were significant differences in total microbiome between the two subtypes. It is worth noting that subtype C2 had a higher observed species index and Shannon index (*t*-test, *p* < 0.001, Figure 3A). Beta diversity analysis further confirmed significant community compositional dissimilarity between the subtypes (PERMANOVA test, *p* < 0.001, Figure 3B). There were also significant differences in microbial diversities between subtype C1 and subtype C2 in the four kingdoms (Appendix A).

We then analyzed the sample distribution of the two subtypes and their efficacy in ICI therapy. The results showed that there were significant differences in the distribution of responders and non-responders as well as the distribution of cases with PFS > 6 months between the two subtypes (*p* < 0.01, Figure 3C). In addition, survival analysis was conducted based on the existing data of overall survival in months (OS_months) and death events. The results showed that subtype C2 exhibited better survival outcomes (*p* < 0.01, Figure 3D). The samples included for subtype C1 and subtype C2 are uniform in terms of sampling timepoints (sampling timepoint) and processing time (SPT) (Appendix A). To further explore potential associations, we analyzed the correlation between subtype classification and patients’ characteristics, including gender, age, BMI, antibiotic use, and therapy. Interestingly, the subtyping result was significantly correlated with therapy and antibiotic use (*p* < 0.01 and *p* < 0.05, Appendix A), while it was not related to gender, age, or BMI (*p* > 0.05, Appendix A).

Furthermore, analysis of the differences in the multi-kingdom microbiome within the subtypes revealed minimal overlap in the species that distinguished responders from non-responders between the two subtypes (Figure 3E and Appendix A). These results indicate that there are significant differences in microbial diversity between the subtypes based on the WSNF method, and these differences are closely related to the outcomes of ICI therapy.

### 3.3. Identification of Distinct Microbial Taxa and Their Effectiveness in Predicting ICI Therapy Outcomes

After comprehensive filtering, we successfully identified 1228 genera, revealing pronounced multi-kingdom compositional differences between subtypes C1 and C2. In the training dataset (randomly sampled 70% of subtypes C1 and C2), we meticulously screened differential species by DESeq2 across two clusters, encompassing 775 bacteria, 51 archaea, 25 fungi, and 79 viruses.

Of all four kingdoms, the bacterial community exhibited the most substantial disparities. Subtype C2 was predominantly characterized by genera such as *Baderoides*, *Enterocloster*, *Parabacteroides*, and *Acidamihococcus*, while subtype C1 was primarily represented by *Denitrobacterium*, *Arabia*, and *Gordonibacter*. A similar pattern was observed in the fungal composition, with *Saccharomyces* and *Kluyveromyces* dominating in subtype C2. The archaeal profile revealed notable differences in methane-producing genera, with *Palaeococcus, Natronorubrum,* and *Sulfurisphaera* more prevalent in subtype C2, and *Methanosphaera* more abundant in subtype C1. The viral community also exhibited multiple significantly differentiated species (Figure 4A). Following a comprehensive analysis of log_2_FC across the different kingdoms, we ultimately selected 32 species as model organisms (7 archaeal, 12 bacterial, 7 fungal, and 6 viral). Subsequent validation utilized various machine learning algorithms to assess the model’s discriminative capabilities for subtype C1/subtype C2 classification within a test set (randomly sampled 30% of subtype C1 and C2). Consistently, the multi-kingdom model demonstrated a trend of superior predictive performance compared to single-domain models (Figure 4B).

### 3.4. Co-Occurrence Networks of Gut Microbiota Reveal Rich Diversity in Subtype C2

In subtype C2, the correlation coefficients for various microbial combinations are more broadly distributed compared to subtype C1 (Appendix A). In the strong correlation section of the B-B combination where ∣ρ∣ > 0.5, subtype C2 reached 21068 (7.02%), while subtype C1 only had 8468 (2.82%). These findings indicate a stronger positive co-occurrence pattern in subtype C2. To further explore microbial interactions, we constructed co-occurrence networks of gut microbiota for these two subtypes. Genus level SparCC correlations were defined as weights within the network, and the Fruchterman–Reingold algorithm was employed to generate the microbiome co-occurrence network. Interestingly, the total number of microbial species and interactions in subtype C2 was significantly higher than that in subtype C1, illustrating a more robust and diverse gut microbiome community (Figure 5A).

Given the absolute dominance of bacterial and fungal communities in the co-occurrence networks, we focused more on the differences in the correlation patterns of bacteria and fungi within the subtype networks. Compared to subtype C1, *Bacteroides* in subtype C2 exhibit more positive correlation patterns (nodes: 12 in C2 vs. 9 in C1) (Figure 5B). In contrast, *Komagataella* show more positive correlation patterns in subtype C1 (nodes: 11 in C1 vs. 9 in C2) (Appendix A). Thus, *Bacteroides* and *Komagataella* are the core genera in the co-occurrence networks of the subtypes.

## 4. Discussion

The homeostasis of gut microbiota is essential for effective ICI therapy, yet most studies have concentrated on the influence of bacteria in this process. As a representative species, *Akkermansia muciniphila* has been shown to restore the efficacy of PD-1 blockade in an interleukin-12-dependent manner by increasing the recruitment of CCR9^+^CXCR3^+^CD4^+^ T cells into the tumor microenvironment [22]. However, there is a growing recognition of the complex interactions between the diverse microbial communities in the gut, including cross-metabolism and other mechanisms. Consequently, to assess the characteristic microbiome of patients undergoing ICI therapy, we utilized the WSNF method to systematically characterize the signature of multi-kingdom microbiome in the gut. Furthermore, in this study, machine learning approaches have identified that the combined multi-kingdom model exhibited superior predictive accuracy compared to any individual single-domain model. This discovery highlights the crucial role of the multi-kingdom microbiome as biomarkers for assessing the efficacy of ICI therapy.

The gut microbiome functions in cancer through various mechanisms, including enhanced antigenicity in microbe–tumor cross-reactivity, microbial-derived metabolic modulation, and innate immune regulation [29]. For instance, Fusobacterium nucleatum could modulate tumor progression by affecting the butyrate metabolism and the autophagy of cancer cells [30,31]. In this study, we identified that the abundance of gut *Bacteroides*, *Parabacteroides*, *Veillonella*, and *Saccharomyces* were elevated in the patients who achieved favorable outcomes with ICI therapy. One previous study demonstrated that *Bacteroides fragilis* can trigger dendritic cell maturation and stimulate IL-12-dependent Th1 cell infiltration, thereby enhancing the anti-tumor effects of anti-CTLA-4 therapy [32]. Additionally, *Bacteroides fragilis* can induce macrophage polarization towards the M1 phenotype and upregulate the expression of CD80 and CD86 on these cells, thereby promoting innate immunity [33]. Furthermore, a recent study demonstrated that *Parabacteroides distasonis* can support ICI therapy-mediated anti-tumor immunity by stimulating the production of CD4^+^ and CD8^+^ T cells within the tumor microenvironment [34]. Notably, *Malassezia* exhibited an increased abundance in the patients obtaining unfavorable therapeutic outcomes with ICI therapy. *Malassezia* infection has been reported to activate mannose-binding lectin, triggering a complement cascade, particularly promoting the production of the oncogenic C3 complement, which facilitates tumor growth [35]. Otherwise, indole compounds produced by *Malassezia* activate the aryl hydrocarbon receptor in host cells, thereby inducing a range of immunosuppressive effects [36]. In this study, beneficial genera such as *Bacteroides* and *Kluyveromyces* have been associated with enhanced responses to ICI therapy, yet the exact biological processes through which they exert their effects need further exploration. In the future, targeting the mechanisms of cancer progression mediated by an identified muti-kingdom microbiome for the optimization of conventional ICI therapy represents a highly promising strategy.

A stable gut ecosystem characterized by high microbial diversity and interactions is closely associated with the host’s immune homeostasis. Previous studies have demonstrated a close relationship between microbial diversity and treatment outcomes, recurrence, and metastasis in cancer. A multi-cohort meta-analysis focusing on CRC revealed a significant reduction in gut microbial diversity in CRC patients compared to healthy controls [37]. Interestingly, similar alterations in gut microbiota diversity have been observed in more distantly located cancer types [38,39]. We also observed that groups with better efficacy of ICI therapy exhibited higher gut microbial diversity through this cross-cohort and multi-cancer analysis [40]. This may signify a more expansive microbial antigen reservoir and a more diverse metabolite spectrum. Such diversity could systematically augment antitumor immunity through enhanced antigen presentation, optimized T-cell activation and functionality, and the generation of immunoregulatory metabolites (such as short-chain fatty acids) [41], thereby improving the therapeutic efficacy of ICI therapy. Moreover, the more intricate microbial co-occurrence networks observed in patients with favorable prognoses in this study may reflect a functionally stable and synergistic microbial ecosystem. This stable network structure potentially contributes to maintaining gut barrier integrity, mitigating harmful bacterial translocation, and ensuring continuous output of critical immunoregulatory functions, collectively creating an immune microenvironment conducive to ICI therapy. Conversely, reduced diversity and sparse microbial networks may precipitate the loss of functional redundancy, ecosystem dysbiosis, and diminished immunoregulatory capacity, ultimately constraining the responsive potential of ICI therapy.

Many previous studies have revealed that the characteristics of the gut microbiome are closely related to the efficacy of ICI therapy in cancer. A study on the gut microbiome in various rare cancers found that the predictive performance based on bacterial characteristics achieved an AUC of 0.73, while *Ruminococcaceae*, *Oscillospiraceae*, and *Lachnospiraceae* made the most significant contributions to cancer prediction performance [42]. Another study identified a group of microbial species associated with the response to ICI therapy in patients with metastatic melanoma, including *Bifidobacterium longum*, *Collinsella aerofaciens*, and *Enterococcus faecium* [19]. However, the majority of these studies are retrospective, highlighting the urgent need for more prospective studies and clinical trials focusing on gut multi-kingdom microbiome biomarkers. This would further validate the strong correlation and causal relationship between the identified microbes and the response to ICI therapy.

Microbiota-based interventions presents a promising avenue for enhancing ICI therapy. Various strategies, including antibiotic treatments aimed at eradicating pathogens, have been proposed. However, exposure to antibiotics can disrupt the homeostasis of the gut microbiota, resulting in reduced microbial diversity and abundance, which in turn impact the immune system and ultimately diminish the efficacy of ICI therapy. A meta-analysis encompassing 2740 patients receiving ICI therapy demonstrated that the use of antibiotics is significantly associated with reduced overall survival and progression-free survival across multiple cancer types [43]. Another animal study found that following antibiotic treatment, the gut of mice became heavily colonized by *Enterocloster*, leading to downregulation of MAdCAM-1 expression. This resulted in the egress of immunosuppressive gut-homing T cells (α4β7^+^CD4^+^ T cells) from the intestines to the tumor-draining lymph nodes, thereby facilitating tumor immune escape and diminishing the efficacy of ICI therapy [44]. This highlights the importance of developing more nuanced strategies that selectively target pathogenic species while preserving or enriching beneficial microbial communities. One promising alternative is FMT. It is noteworthy that a Phase I clinical trial on FMT reported that patients who were unresponsive to ICI therapy exhibited therapeutic effects after undergoing FMT; in addition, the findings indicated that higher levels of *Veillonellaceae* and lower levels of *Bifidobacterium* are closely associated with improved efficacy of ICI therapy [45]. Another clinical trial found that FMT significantly improved the efficacy of anti-PD-1 inhibitors in patients with unresectable or metastatic solid tumors who were otherwise refractory to anti-PD-1 therapy, identifying *Prevotella merdae* as a key microorganism responsible for enhancing the efficacy [46]. Furthermore, a meta-analysis revealed that the objective response rate of FMT combined with immune ICI therapy at 43% is significantly higher than the 20–40% response rate reported for monotherapy in previous studies [47]. Furthermore, the use of probiotics and prebiotics represents another viable strategy for modulating the microbiome. One randomized trial in metastatic renal-cell carcinoma demonstrated that the probiotic strain *Clostridium butyricum* (CBM588) significantly prolongs progression-free survival and increases objective response rates when combined with nivolumab plus ipilimumab [48]. On the prebiotic side, dietary supplementation with inulin-type fructans enhances anti-PD-1 efficacy in murine tumor models by expanding *Faecalibacterium prausnitzii* and elevating fecal butyrate, which promotes intratumoral IFN-γ^+^CD8^+^ T cell infiltration and reduces exhausted PD-1^+^TIM-3^+^ T cells [49]. Together, these approaches may facilitate the maintenance of a balanced microbiome, ultimately contributing to improved immune function and therapeutic outcomes in patients undergoing ICI therapy.

Grasping the influence of multi-kingdom microbiomes on ICI therapy presents an important opportunity to enhance cancer treatment protocols. Integrating microbiome research into conventional cancer therapies could enable the development of more personalized and effective treatment strategies, ultimately resulting in improved outcomes for cancer patients.

## 5. Conclusions

This study provides a comprehensive analysis of the gut microbiome characteristics of patients undergoing ICI therapy. Through multi-kingdom microbial clustering using the WSNF method, we identified two distinct patient subtypes with significant differences in microbial diversity and therapy outcomes. Our findings highlight the importance of microbial diversity in predicting the efficacy of ICI therapy and suggest that the multi-kingdom microbiome could serve as a potential biomarker for treatment response. The identification of specific microbial taxa associated with favorable outcomes and the construction of co-occurrence networks further underscore the complex interactions within the gut microbiome and their impact on ICI therapy. This meta-analysis systematically reveals the critical regulatory role of the multi-kingdom microbiome in ICI therapy and offers a novel perspective for personalized therapy.

## Figures and Tables

**Figure 2 microorganisms-13-02595-f002:**
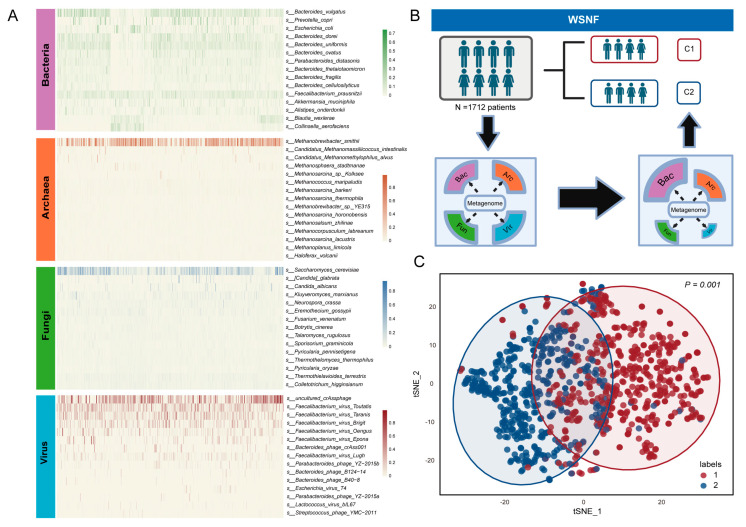
Multi-kingdom microbial clustering using WSNF method. (**A**) The heatmap shows the top 15 abundant species in the four clades of bacteria, archaea, fungi, and viruses, with color intensity representing the relative abundance of each species in the samples. (**B**) Flowchart of the study design using WSNF clustering method. Based on the WGS data, the samples were divided into two subtypes, C1 and C2. (**C**) The t-distributed stochastic neighbor-embedding (t-SNE) visualization method was used to display the subtyping results (*p* < 0.01, and the circles show the 95% confidence interval).

**Figure 3 microorganisms-13-02595-f003:**
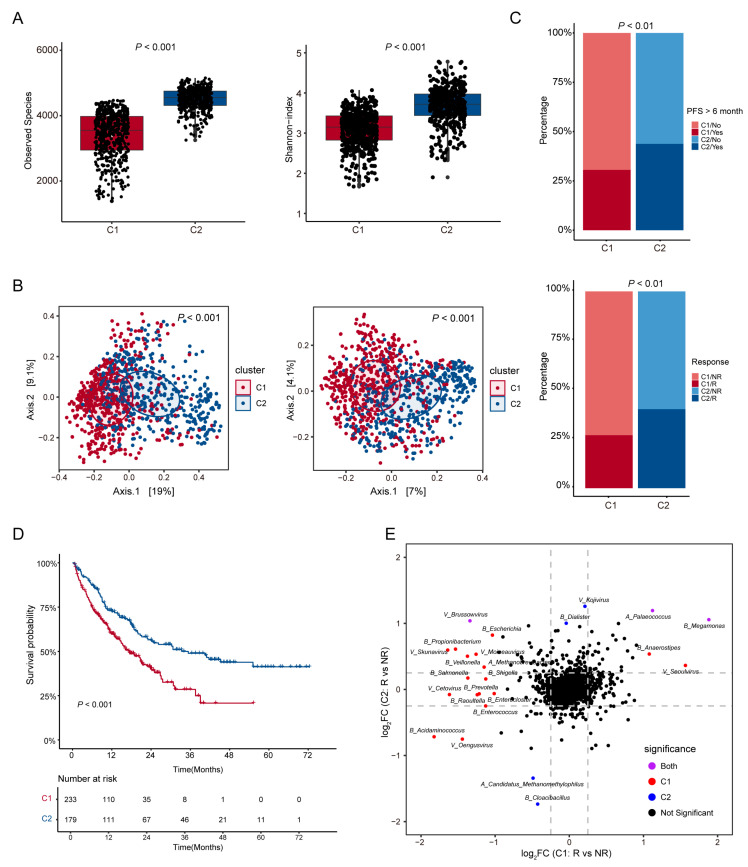
Analysis of microbial diversity and its correlation with ICI therapy outcomes between two subtypes. (**A**) The differences in alpha diversity between the two subtypes were examined, incorporating the observed species and Shannon index. The *t*-test was used to determine the differences. The two colors represent different clusters on the x-axis. The dots represent samples and their positions correspond to the observed species values or Shannon index values for those samples. (**B**) Principal coordinate analysis (PCoA) based on Bray–Curtis distance and Jaccard distance was conducted. The PERMANOVA method was used to determine the differences. (**C**) The distribution of responders (R) and non-responders (NR), as well as the distribution of patients with PFS > 6 months among different subtypes were analyzed, and chi-square test was used to determine the differences. (**D**) Survival analysis was performed based on the available data of overall survival in months and death events. (**E**) Pattern species for distinguishing responders (R) from non-responders (NR) among two subtypes.

**Figure 4 microorganisms-13-02595-f004:**
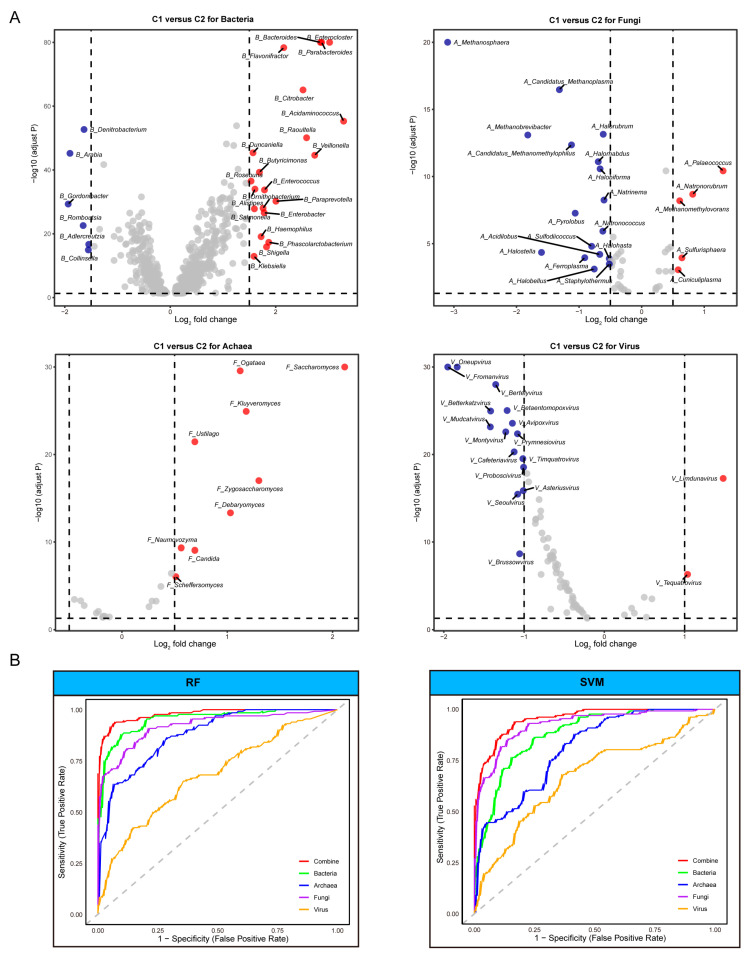
Identification of distinct microbial taxa and the effectiveness in predicting ICI therapy outcomes. (**A**) Analysis of differences at the genus level between different kingdoms. The dashed line represents the threshold of log_2_FC. Red dots indicate significantly different species in subtype C1 (*p* < 0.05), blue dots are significant in subtype C2 and gray dots are not significant between the two clusters. (**B**) Machine learning based on model species to predict the ROC curve differentiating ICI efficacy, including multi-kingdom combined models and single-kingdom models. This dash line represents the point at which the true positive rate (TPR) and false positive rate (FPR) are equal in machine learning model.

**Figure 5 microorganisms-13-02595-f005:**
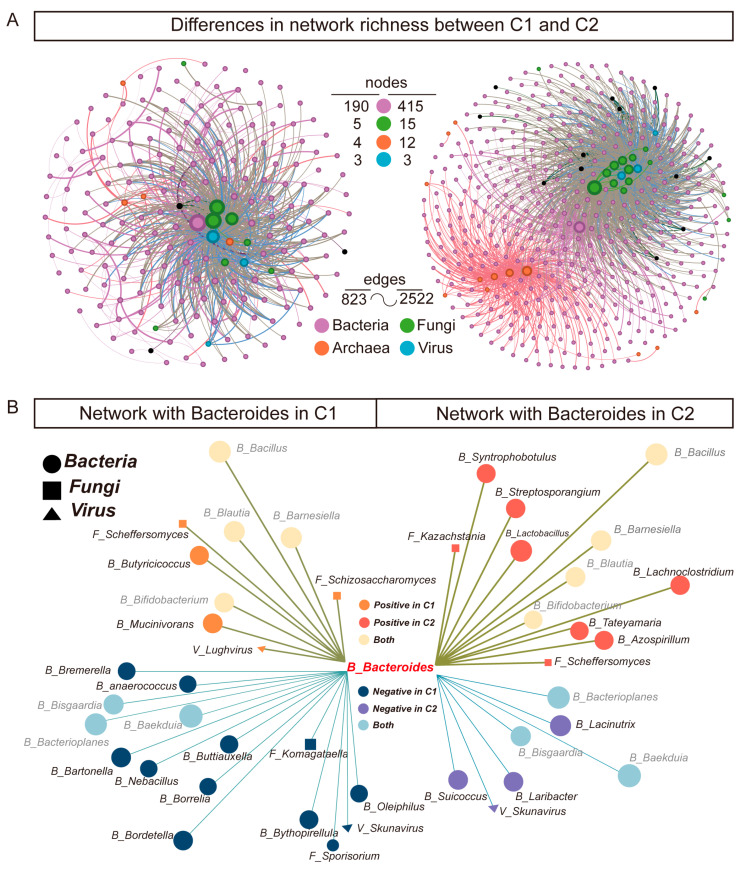
Co-occurrence analysis of the multi-kingdom network between subtype C1 and subtype C2. (**A**) Co-occurrence network in subtypes, microorganisms are defined as nodes, with the magnitude of SparCC correlations being proportionate to the width of the edges between nodes. The Fruchterman–Reingold algorithm was utilized for the layout of nodes and edges. Node size is directly proportional to the degree. (**B**) Network with *Bacteroides* in subtypes. The left side of the image shows the correlation of *Bacteroides* in subtype C1, while the right side shows the correlation of *Bacteroides* in subtype C2.

**Table 1 microorganisms-13-02595-t001:** Comprehensive characteristics of cohorts included in this study.

Cohort	Sequencing Type	Country	Cancer Type	Treatment	Patients Included (Responder/Non_Responder)
PRJNA762360	Metagenomic sequencing	USA	Melanoma	Anti-PD-1 blockade	63 (38/25)
PRJEB43119	Metagenomic sequencing	Netherlands, Spain, UK	Melanoma	Combination of anti-PD-1 and anti-CTLA-4 blockade	165 (100/65)
PRJNA615114	Metagenomic sequencing	China	Gastrointestinal cancer	Anti PD-1/PD-L1 blockade	114 (70/44)
PRJNA751792	Metagenomic sequencing	France	NSCLC	Anti-PD-1 blockade	337 (75/262)
PRJNA541981	Metagenomic sequencing	USA	Melanoma	Anti-PD-1 blockade/anti-CTLA4 blockade/combination of anti-PD-1 and antiCTLA-4 blockade	113 (58/55)
PRJEB54704	Metagenomic sequencing	Germany, USA	B cell lymphoma	CD19-targeted CAR-T cell immunotherapies	351 (149/202)
PRJEB22893	Metagenomic sequencing	USA	Melanoma	Anti-PD-1 blockade	25 (14/11)
PRJEB22863	Metagenomic sequencing	France	NSCLC, RCC	Anti-PD-1/PD-L1 blockade	217 (36/181)
PRJNA399742	Metagenomic sequencing	USA	Melanoma	Majority with anti-PD-1 blockade, a small portion with anti CTLA-4 blockade	172 (11/18)
PRJNA397906	Metagenomic sequencing	USA	Melanoma	Anti-PD-1 blockade and other immune checkpoint blockades	44 (19/20)

## Data Availability

The raw metagenomic data can be obtained from the SRA (https://www.ncbi.nlm.nih.gov/SRA) and the European Nucleotide Archive (https://www.ebi.ac.uk/ena/) under the accession numbers (PRJNA762360, PRJNA615114, PRJNA541981, PRJEB22893, PRJNA399742, PRJNA77029, PRJNA751792, PRJEB22863, PRJEB54704, and PRJEB43119). The code and scripts are available on GitHub (https://github.com/20050515/cmyd1, accessed on 25 June 2025). The customized code was written in R (version.4.3.1).

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
