# Peer review of "Multi-Kingdom Gut Microbiome Interaction Characteristics Predict Immune Checkpoint Inhibitor Efficacy Across Pan-Cancer Cohorts"

_microorganisms, 2025, doi:10.3390/microorganisms13112595_

Round 1
Reviewer 1 Report
Comments and Suggestions for Authors
In the article authors analyzes:
- Gut microbiome characteristics in patients receiving immune checkpoint inhibitors (ICIs)
- Multi-kingdom microbial clustering using WSNF method
- Two distinct patient subtypes with different microbial diversity and clinical outcomes
- Potential biomarkers for treatment response
- Complex interactions within the gut microbiome affecting immunotherapy
The figures in this work are very good - they are clear, readable and perfectly fit the presented study. All graphic elements are presented transparently, which facilitates understanding of the presented results and methodology. The quality of data visualization is at a high level and effectively supports the scientific narrative of the publication.
- Materials and Methods. Study Cohorts
Several critical methodological questions require clarification regarding the study design. What specific inclusion and exclusion criteria were applied for selecting the 10 geographical populations from available metagenomic databases, and how was sample representativeness ensured across different cancer types and regions? Additionally, what standardization procedures were implemented to harmonize datasets from different sequencing platforms and studies, particularly regarding batch effect correction and quality control metrics? The technical processing methodology also needs elaboration, specifically which reference genomes and bioinformatics pipelines were used for shotgun metagenomic alignment, taxonomic classification, and normalization of sequencing depth differences. Finally, details regarding the metadata curation framework, including standardization protocols for clinical variables and validation procedures for ensuring data accuracy across the multi-center cohort, would strengthen the methodological transparency and reproducibility of this multi-kingdom microbiome analysis.
Reviewer 2 Report
Comments and Suggestions for Authors
Overall I think this paper is strong and incredibly interesting. The authors methods are sound and the work is of interest to bioinformaticians and microbiome researchers. Very well organized with good headings. Just a few minor technical comments.
- Line 81: Good number of samples à Increase in experimental power
- Lines 84-86: Good diversity in locations for dataset. Glad the authors input all of the PRJNA #s.
- Lines 94: Please put commas here.
- Section 2.5/2.6: Can the authors please elaborate a bit here on why these methods are important. R methods are well describe though.
- Lines 161, 191, 254, 256, 258, 461, 463, 465, 467, 469: No figure descriptions for all figures. Labeled as “Caption”. Please also label supp figures.
- Figure 1: Excellent figure. As is figure 2.
- Lines 254, 256: Figures 3 and 4 are not mentioned or referenced in the article. (Unless I accidently missed it)
- Lines 300-302: Elaborate on future directions.
- Discussion: overall good but a bit brief. Conclusion is fine.
- Lines 337-341: Data and code used are available to the public. Good.
- Citations: make sense.
Reviewer 3 Report
Comments and Suggestions for Authors
I read this promising manuscript with great interest; however, I have some comments to raise to the authors:
(1) The sample size appears to vary throughout the manuscript: it initially refers to 1,723 patients across 9 cohorts, then changes to 1,713 across 10 cohorts. This should be clarified more precisely.
(2) No transformation is applied to address compositionality beyond Hellinger.
(3) Alpha diversity is assessed using a t-test on data that are themselves used to generate the clusters, which introduces the risk of circular significance. It would be advisable to consider permutations or bootstrap tests before clustering.
(4) The survival analyses appear unadjusted for cancer type, grouping together extremely heterogeneous cancers.
(5) Figure titles are missing.
(6) There are many line-break errors and multiple grammatical mistakes; careful proofreading of the manuscript is needed (e.g., “soild tumors”).
(7) The manuscript states that 33 informative genera were selected from the full set of 965 samples using significance testing or variable importance, and then the same sample was split into a training set (70%) and a test set (30%), and the random forest classifier was trained on those 33 features. This results in data leakage because information from the test set is indirectly seen during the feature selection phase. As a result, the features are optimised not only to distinguish the classes in the training data but also for those particularities (including noise) present in the evaluation data. This leads to an unjustified advantage, causing AUCs to be overestimated and poorly generalisable. Techniques such as nested cross-validation are missing to mitigate this.
(8) The manuscript states that the B-B network includes 82,322 positive edges after filtering out correlations with |ρ| < 0.5. With 775 bacterial genera, the maximum number of undirected pairs is 299,925 (775*774/2). Therefore, the authors are claiming that approximately 27% of all possible pairs simultaneously exceed the |ρ| > 0.5 threshold and (presumably) an FDR < 0.05. With ~965 samples, such a high density is statistically and biologically unlikely… This point should be further discussed.
Please proofread the manuscript.
Round 2
Reviewer 3 Report
Comments and Suggestions for Authors
The manuscript has been revised. No further comments to the authors.